# Geonews: Timely Geoscience Educational YouTube Videos about Recent Geologic Events

Ning Wang[1], Zachary Clowdus[1], Alessandra Sealander[1] and Robert J Stern[1]

[1]Department of Geosciences, University of Texas at Dallas, Richardson, 75080, USA.

*Correspondence to*: Ning Wang (Ning.Wang@utdallas.edu)

**Abstract.** Geologic events like volcanic eruptions, earthquakes, and tsunamis hurt nearby people and stimulate the curiosity of people farther away, thus providing opportunities to engage the public to be more interested to learn about Earth processes. Geoscientists are increasingly using social media such as Twitter to explain to the public what caused these events and videos provide an especially vivid way to reach this

audience. However, it is still unclear how to create, evaluate and disseminate videos on timely natural events to communicate geosciences. To address this challenge and opportunity, we analyzed the impact of 33 short geoscience educational (GeoEd) videos that we created and posted on YouTube between 2018 and 2020. These include 12 videos on timely geologic events (Geonews videos) and 21 videos that are not specially about timely geologic topics (General GeoEd videos), all of which were similarly advertised and

have similar lengths. By comparing the performance of the Geonews and General GeoEd videos, we conclude: 1) The YouTube audience is consistently interested in Geonews videos but some General GeoEd videos are more popular; 2) Geonews videos may trigger more meaningful dialogues than General GeoEd videos, especially for local audiences; 3) The 'golden period' of Geonews videos engaging YouTube audiences is within 3 weeks after posting; 4) The Geonews audience tends to be younger and more diverse

than the General GeoEd video audience; 5) Creating Geonews videos can be a promising strategy for geoscientists to engage public audiences on YouTube-like social media.

## 1. Introduction

Effectively communicating science to the public is challenging (Allum et al., 2008; Dyer, 2018; Bartel and Bohon, 2019; Greussing et al., 2020) but news about natural hazard events like earthquakes, tsunamis, and

volcanic eruptions attracts people's attention and create opportunities for two-way dialogues about geosciences (Falk and Dierking, 2010; Tong, 2014; Barrett et al., 2014; Illingworth et al., 2018). Some research suggests that discussing the science behind such events soon after they occur on message-based social media, such as Twitter, can engage the public who want to learn more (e.g. Rosenbaum and Culshaw, 2003; Veil et al., 2011; Drake et al., 2013; Shiffman, 2017; Takahashi et al., 2015; Lacassin et al.,

2020; Wibisono et al., 2020). However, few studies have tested if the same strategy can also be successfully applied to videos posted on YouTube (Schäfer, 2012; NAS, 2017). This work addresses two questions: First, would videos posted on YouTube about Earth events and processes also stimulate the public to be more interested in these? Second, are YouTube users more interested in timely events-based

geoscience educational videos (herein referred as to 'GeoEd videos') relative to videos that are unrelated to recent events in the news?

Social science provides the fundamental theories of how to effectively communicate geoscience to the public (Nisbet et al., 2010; Illingworth et al., 2015). With more and more evidence against the early one-way expert-to-public knowledge-transfer model (known as 'information deficit model'), researchers increasingly suggest that it is important to value 'lay local' knowledge to stimulate dialogues and better communicate science to the public (Irwin and Michael, 2003; Allum et al., 2008; Siersdorfer et al., 2010; Illingworth et al., 2015; Stewart and Lewis, 2017; Illingworth, 2017). Also, although meta-analysis on overall public knowledge and attitude about science shows a weak positive relationship, results varied for different subjects (Allum et al., 2008). Geoscience has three unique features regarding communicating with public. First, understanding how complex Earth systems operate is complicated because many Earth processes cannot be directly observed: They occur deep in the Earth and/or over unimaginably long timescales (Singer et al., 2012; Willis et al., 2021; Mosher and Keane, 2021). Dealing with geoscientific information can easily cause a high cognitive load (Arthur, 2018). Therefore, communicating geoscience to the public should strive to reduce cognitive load. Secondly, different geoscience aspects are more relevant to some places than others (King, 2008), for example Californians are more interested in earthquakes than hurricanes and Floridians are more interested in hurricanes than earthquakes. Different places also have different communities sharing local cultures and beliefs (Michael, 2009), so that taking advantage of local context and geological events is especially important for public engagement (Takahashi et al., 2015; Semken et al., 2017). Thirdly, geoscience topics often concern dynamic and complex systems, involving much uncertainty and chaos (Manduca and Kastens, 2012; Stillings, 2012). This makes visual storytelling, multimedia and two-ways conversations (between the public and experts) even more important (Nisbet et al., 2010; Mosher et al., 2014; Urban and Falvo, 2016; Mosher and Keane, 2021). Lastly, explaining Earth science concepts also requires understanding different components of an Earth system and how these interact (Forster and Freeborough, 2006; Bobek and Tversky, 2016; Lacchia et al, 2020). The challenge of explaining this complexity encourages more geoscientists to explore using social media for communicating geosciences to the public. We need to learn more about how to best use different types of social media to communicate geoscience issues to them (Schäfer, 2012; Dunn, 2013; Illingworth et al., 2018).

Videos have special advantages for communicating geoscience to the public and beginning students compared to words alone or words and static figures combined (Nisbet et al., 2010; Wiggen and McDonnell, 2017; Littrell et al., 2020). Most difficulties of communicating geoscience mentioned above can be overcome with videos and animations (Wijnker et al., 2019; Ploetzner et al., 2020) and by integrating psychological designs into repeatable educational units (Goldberg et al., 2019; Greussing et al., 2020; Mayer, 2021). Moreover, research has shown that YouTube videos can involve large numbers of people to be more interested in important geoscience issues such as climate change (Zavestoski et al., 2006; Askanius and Uldam, 2011; Krauss et al., 2012; Stewart and Nield, 2013; Van Loon et al., 2020). Videos

also have the advantage of being organizable into YouTube channels where they are more easily found to be used for teaching and learning in diverse environments (Welbourne and Grant, 2016; Maynard, 2021). Furthermore, YouTube provides a 'comments' function which makes dialogue possible. Therefore, it is valuable to understand if and how timely, short videos about geologic events in the news posted on YouTube can reach the public and trigger meaningful dialogue.

In this study, we analyzed the performance of 33 GeoEd videos (all less than 6 mins with elaborated editing) that we posted on YouTube in 2018 and 2020, paying attention to who was interested in these and for how long as well as what dialogue occurred in the comments. These include 12 timely videos about natural events in the news ('Geonews videos') and 21 GeoEd videos about processes that are not time-sensitive because they are not about something that just happened ('General GeoEd videos'). Geonews

videos are mostly published about 2 weeks after the event occurred. General GeoEd videos aims to explain some geological concepts or phenomenon and do not utilize timely events to engage the audiences; These are created with less urgency and take longer to make. By comparing the performance of Geonews and General GeoEd videos, we explore the advantages and limitations of the Geonews format. Using data from YouTube Analytics and Comments, we can evaluate audience engagement with these two types of videos

that we made and posted in 2018 and 2020 (2019 was excluded because no Geonews videos were posted in 2019).

    This study (1) introduces how we design Geonews videos; (2) compares the performance and audience features of Geonews and General GeoEd videos on YouTube; and (3) explores how and why Geonews videos engages a different group of viewers. Our results indicate that using Geonews-like videos to explain

what, where, and why geologic events happen is a useful strategy for engaging diverse YouTube users.

## 2. Geologic Events and Geoscientific Outreach

Using geologic events to interest and teach people has been long discussed (Vitek and Berta, 1982). Most research about communicating natural hazards to the public focuses on preparing for potential disasters, emphasizing what people should do during a geologic disaster and how to be resilient afterwards

(Rosenbaum and Culshaw, 2003; Forster and Freeborough, 2006; Ickert and Stewart, 2016; Kelly and Ronan, 2018). With the development of the internet, computers and smartphones, social media is increasingly acknowledged as a key tool for the communication and education activities of emergency agencies. More and more geoscientists highlight the importance and effectiveness of using these new tools to reach and teach the public and beginning students after a natural hazard event happens (Bartel and

Bohon, 2019; Barton et al., 2020; Lacassin, et al., 2020). Most studies document effective and ineffective uses of social media in crises, focusing on topics such as fast communication, accuracy, credibility, uncertainty, and communicating broadly (Freberg et al., 2013). Using social media as disaster resilience communication tools in addition to traditional engagement and education activities is well studied (Dufty, 2011; Veil et al., 2011; Freberg and Palenchar, 2013; Lundgren and McMakin, 2013).

The need to enhance public perception of geology and natural hazards, educate them about the Earth, and recruit geoscience students continues to increase (Rosenbaum and Culshaw, 2003). As a result, geoscientists increasingly apply an event-based method in a cultural context to discuss geologic events and natural hazards on social media (Illingworth, 2018; Fallou and Bossu, 2019). There are several popular social media platforms that are available but probably the most studied and used is Twitter. Considering the

need to respond as fast as possible to disasters, this is understandable. Twitter messages are short and very interactive. Twitter allows geoscientists to provide useful information almost immediately after an event (Hicks, 2019). Writing text and posting "point-and-click" photos and camera-recordings of an event is easier and faster than creating GeoEd videos which must provide context, consider educational effects, and require more time.

Researchers have used a case-based and descriptive way to study the effects of using Twitter to communicate to the public about geologic events, showing that Twitter can gain the attention and inform the public quickly (Rosenbaum and Culshaw, 2003; Lomax et al., 2015). These studies find that such events allow geoscientists to communicate pertinent scientific information to the public but many aspects are not well explained by Twitter and similar social media (Mossoux et al., 2016; Lacassin et al., 2020).

The need for jargon-free explanation with coordinated graphical elements is not met with these social media platforms. These shortcomings can be overcome by making short videos that provide context and visual clues with embedded educational designs and input from more than one person (including experts). Such videos, if available soon after the event, can powerfully complement "on the spot" Twitter and similar social media posts. Well-crafted, short videos about a newsworthy event can be engaging and can possibly

better manage cognitive load of the public than can texts, pictures, or unedited videos without educational considerations. In addition, videos can be embedded into websites and other social media like Facebook and Twitter (Moloney and Unger, 2014).

Edited videos play an increasingly important role in informal education and are popular worldwide (Thomson et al., 2014; Welbourne and Grant, 2015; Wijnker et al., 2019; Vega and Robb, 2019). YouTube

is the main platform for these and has about two billion users every month (Welbourne and Grant, 2015; YouTube, 2021). This audience uses YouTube videos for much more than entertainment; about half of YouTube adult use is for learning (Smith et al. 2018; Allgaier, 2020). YouTube videos can help communicate Earth science to the public because this is not easy (Dyer, 2018). Earth science concepts have many elements that are unfamiliar: They occur in strange lands or under the sea, and involve words and

concepts that are abstract, complex, and confusing (Greussing et al., 2020; Stern et al., 2020). Well-crafted GeoEd videos are especially effective for revealing the meaning of unfamiliar words to the public and explaining abstract and complex geoscience concepts to them (e.g. Banchero et al., 2021; Schmidt-McCormack et al., 2017; Akinbadewa and Sofowora, 2020; Stern et al., 2017 and 2020; Tayne et al, 2021; Wang et al., submitted). However, despite evidence of the power of this approach, there is little known

about the advantages and disadvantages of utilizing YouTube videos about recent geologic events to reach

and teach (Nisbet et al., 2010; Schäfer, 2012; Takahashi et al., 2015). Few have studied the potential of using videos on the internet to explain recent geological events and natural hazards as a way to engage the much larger group of people who do not directly suffer from the event. Also, it is unclear if those who are impacted by an event or know someone directly impacted are better engaged by Geonews-like videos about it.

**3. Geonews Videos**

All UTD Geonews videos are about 3 to 5 mins long and created by geoscience students in the Geoscience Studio at the University of Texas at Dallas (UTD GSS). The GSS team is supervised by Professor Stern and creates all types of short GeoEd videos. A subset of these are assessed in the classroom, especially ones intended for undergraduate classes (Stern et al., 2017; Willis et al., 2021; Wang et al. submitted). Geoscience Studios began in 2016 and we began making Geonews videos in 2018. All Geonews videos have a similar format (Figure 1): 1) Start with a simple introduction of the event, including location and date; 2) Explain pertinent background; and 3) Provide a simple scientific explanation for the event, along with scientific evidence. In some cases, we introduce some relevant basic geoscientific concepts such as normal faults, plate tectonics, or earthquake magnitude. In some cases, we reach out to experts and get their input. All Geonews videos conclude with references and web links where interested viewers can learn more.

The workflow of making a Geonews video begins with: (1) Someone proposes an ongoing or recent event as a topic for a new video to the UTD GSS video production team. (2) Once the UTD GSS team agrees, a production leader volunteers and works with Prof. Stern to collect information, images and videos on the topic. (3) A 360-600 words narrative is written by the production leader and Prof. Stern, setting the length and pace for a 3-5 minute video. (4) The narrative is recorded (the narrator is also a UTD student) and graphics and background music added. (5) Once the video is finalized, it is posted on the UTD GSS YouTube channel and closed captions would be added and corrected. Once this is done, it is advertised to various on-line scientific communities such as the Geological Society of America, the American Geophysical Union, Sigma Xi, and the American Association for the Advancement of Science. These are also advertised on Facebook on our personal accounts and in a Facebook public group "Geoscience Animations and Videos" (279 members as of Oct. 2021). In addition, the growing subscriber base for the UTD GSS YouTube channel (~2270 as of Oct. 2021) is also notified. This procedure allows us to release a Geonews video within about 2 weeks after we begin work.

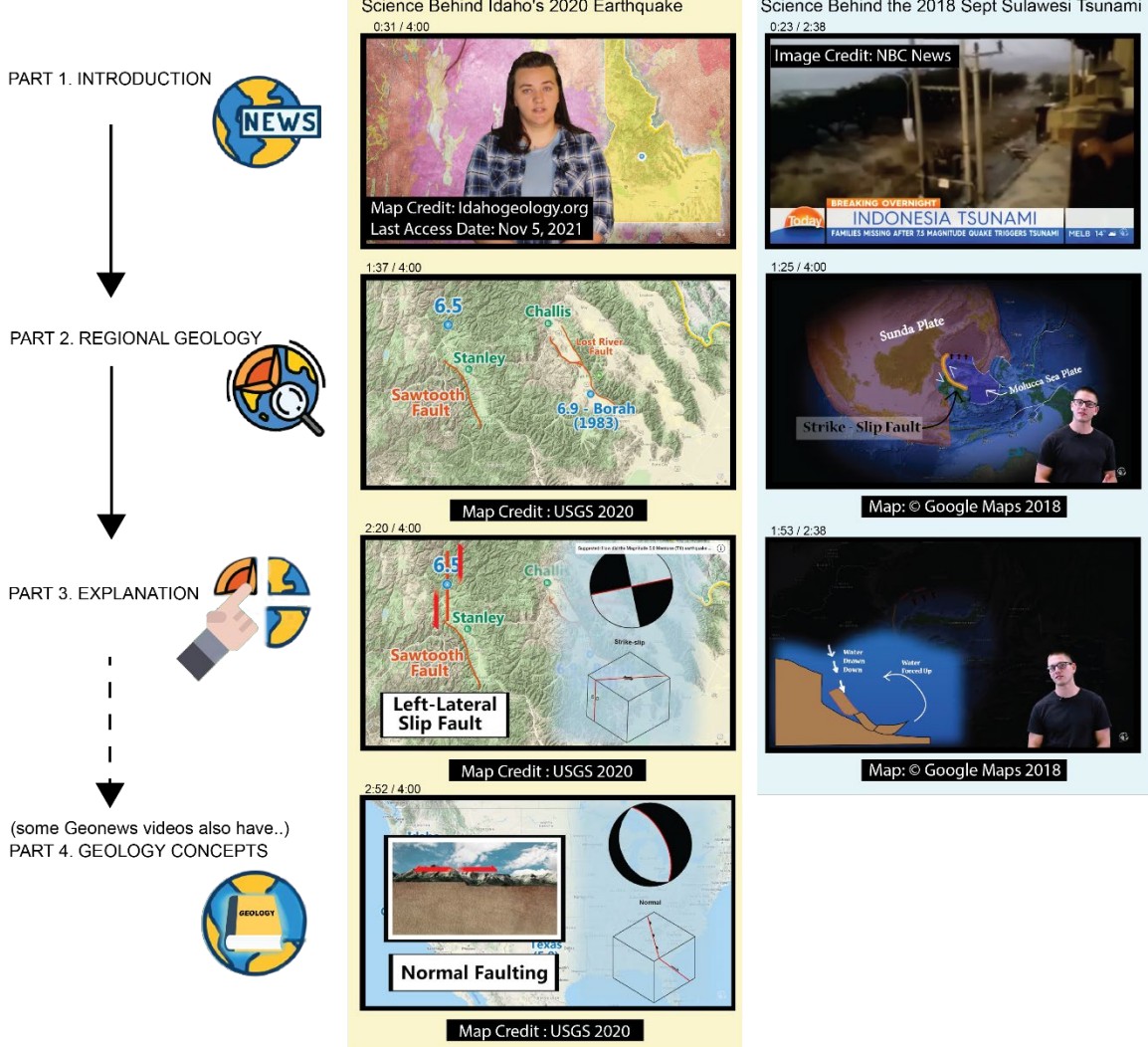

**Figure 1. Design framework of Geonews videos and two examples. Details and links of the two Geonews video examples can be found in Table 1. (Map: © Google Maps 2018; U.S. Geological Survey, earthquake.usgs.gov, 2020; © OpenStreetMap contributors; NBC News Today, 2018, last access: 9 Nov, 2021; Idaho Geological Survey, 2020)**

From our experience, Geonews videos are easier to make than General GeoEd videos for three reasons:

(1) The design is more standardized.

(2) Because the event just happened, a lot of relevant information (especially visual materials) is easy to find. It is easier to find relevant materials by keyword search, and easier to find experts to consult.

(3) Because the video concerns a single event, it is easier to pull together a story and write the narrative.

**Table 1. List of 12 Geonews videos (2018 - 2020)**

| # | Title | Short Description (Location, TYPE*) | Link | Total Length |
|---|-------|-----------------------------------|------|--------------|
| 1 | The Sinabung Volcano Eruption! | Indonesia, VE | https://youtu.be/t0xwiS2mW5k | 2min35sec |
| 2 | Science Behind the Earth Suswa Fissure (Kenya) | Kenya, East Africa, FI | https://youtu.be/sOB7O3yvC4Q | 3min14sec |
| 3 | Science Behind Hawaii Eruption 2018 | Hawaii, US, VE | https://youtu.be/f-Z5d2ZBIro | 4min50secs |
| 4 | Science Behind the 2018 Sept Sulawesi Tsunami | Indonesia, TS | https://youtu.be/1oaI4Mo7V_s | 2min39sec |
| 5 | Taal Volcano Eruption 2020 | Philippines, VE | https://youtu.be/z-iKOBjIiYc | 2min43sec |
| 6 | Science of the Magnitude 5.7 Magna, Utah earthquake | Utah, US, EQ | https://youtu.be/d6R6FTQnR3U | 2min48sec |
| 7 | Science of the Magnitude 5.0 Mentone (TX) earthquake | Texas, US, EQ | https://youtu.be/MfxmvXsIpBI | 3min23sec |
| 8 | Science Behind Idaho's 2020 Earthquake | Idaho, US, EQ | https://youtu.be/s_5YKFR5AMU | 4min1sec |
| 9 | Science Behind Nevada's 2020 Earthquake | Nevada, US, EQ | https://youtu.be/GizueyqNwYQ | 5min |
| 10 | Science Behind Mexico's 2020 Earthquake | Mexico, EQ | https://youtu.be/mIlQqfj8MQY | 4min15sec |
| 11 | Science Behind the 2020 Sparta, North Carolina Earthquake | North Carolina, US, EQ | https://youtu.be/JDz5UDbVGb8 | 3min40sec |
| 12 | Science Behind the 2020 Aegean Sea Earthquake | Turkey and Greek Islands, EQ | https://youtu.be/MMBFY-LahNc | 5min1sec |

(*EQ - Earthquake, VE - Volcano Eruption, TS - Tsunami, FI – Fissure)

## 4. Methods and Materials

To better understand how focusing on timely natural hazard elements affects audience engagement with short videos, we compared Geonews videos with other short GeoEd videos we made that have a different
focus (General GeoEd videos). We use General GeoEd videos as a control to study the effects of Geonews videos. By comparing the performance of Geonews and General GeoEd videos that we created and posted on YouTube in 2018 and 2020, we isolate the effects of timely reporting on natural hazards in engaging the audience. We exclude 2019 GeoEd videos because no Geonews videos were made that year (UTD GSS activities depend heavily on UTD student interest and availability). The two types of videos were posted in
the same years, eliminating engagement differences caused by continuously growing numbers of

subscribers to the UTD GSS channel and our improving video-making skills. In 2018 and 2020, a total of 33 short GeoEd videos were posted on YouTube, including 12 Geonews videos (Table 2A) and 21 General GeoEd videos (Table 2B). In 2018, we posted 4 Geonews and 6 General GeoEd videos, increasing to 8 Geonews and 14 General GeoEd videos in 2020. The topics were chosen based on educational need, event impact, and UTD GSS team interest and availability. Some General GeoEd videos were made as undergraduate class projects. All the videos were reviewed and directed by Prof. Stern and other content experts to ensure accuracy.

All videos followed a similar video-making philosophy and workflow to ensure quality, artistic skills, project duration and dissemination strategies. The average length of the 12 Geonews videos is 3min 41sec (std. dev. = 1min 18sec) and that of the 21 General GeoEd videos is 3min 55sec (std. dev. = 1min 13sec). The range of lengths of Geonews and General GeoEd videos are also similar (from ~2min 30secs to ~5min). Both Geonews and General GeoEd videos were disseminated similarly. These similarities ensure the differences in audience response mostly reflect differences in timeliness: for Geonews videos, a focus on something that just happened, whereas for General GeoEd videos, there was no such focus.

We examined six factors available from YouTube statistics and comments to assess the nature of the audience and its engagement for the two groups of videos (Table 2). For engagement, we examined the number of views, average percentage of video watched (herein referred as to 'average percentage viewed'), like/dislike ratio, as well as analyzing all comments (Azer et al., 2013; Allgaier, 2019; Ozdede and Peker, 2020). Number of views reflect how interested the audience is in the topic: More views indicate more interest. We also compared the two groups over different time periods (15 weeks after video release as well as lifetime performance) to see how important timeliness was. Data was collected from YouTube Analysis. To assess how successfully the video retained audience interest, we also compared the two groups' average percentage viewed. This reflects video quality: higher percentage watched indicates a more engaging video (Guo et al., 2014). In addition, analysis of comments is useful for exploring in greater depth YouTube users' attitudes towards the information presented (Chatzopoulou et al., 2010; Hussain et al., 2018; Dubovi and Tabak, 2020). We analyzed 222 comments as of 10/03/2021 to understand how many meaningful dialogues were triggered. Like/dislike ratio indicates the users' attitudes about each video (Ozdede and Peker, 2020). Lastly, in order to understand audience demographics for the two GeoEd video groups, we also compared their ages and genders in an effort to understand if Geonews and General GeoEd videos engaged different audiences.

Two metrics that could be relevant to engagement are not considered: watching time and average view length. These are related to engagement but since the two groups of videos have very similar average lengths, these two metrics can be approximately represented by views and average percentage viewed.

**Table 2. Details of 12 Geonews videos and General GeoEd videos created in 2018 and 2020\*. (\* as of Oct 03, 21; -us- indicates US related events)**

(A) Geonews Videos:

| # | Video Example | Event Date | Release Date | Intensity (Mw/VEI/TIS) | Interval (Days) | Views * | Average View Percentage | Comments |
|---|---|---|---|---|---|---|---|---|
| 1 | The Feb 2018 Sinabung Volcano Eruption | Feb. 19 2018 | Feb. 27 2018 | VEI 4 Little Damage And Largely Observed | 18 | 2,397 | 68% | 3 |
| 2 | Science Behind the Earth Suswa Fissure (Kenya) | Mar. 27 2018 | Apr. 14 2018 | Little Damage And Largely Observed | 18 | 2,309 | 67% | 1 |
| 3 | Science Behind Hawaii Eruption 2018 -us- | May 06 2018 | May 18 2018 | VEI 0~3 Very Destructive | 12 | 5,001 | 61% | 7 |
| 4 | Science Behind the 2018 Sept Sulawesi Tsunami | Sep. 28 2018 | Oct. 14 2018 | TIS X~XII Very Destructive | 16 | 5,407 | 66% | 8 |
| 5 | Taal Volcano Eruption 2020 | Jan. 12 2020 | Jan. 16 2020 | VEI 4 Little Damage And Largely Observed | 4 | 2,417 | 59% | 0 |
| 6 | Science of the Magnitude 5.7 Magna, Utah earthquake -us- | Mar. 18 2020 | Mar. 29 2020 | Mw 5.7 Frightened All Damage Negligible | 11 | 4,893 | 67% | 3 |
| 7 | Science of the Magnitude 5.0 Mentone (TX) earthquake -us- | Mar. 26 2020 | Apr. 06 2020 | Mw 4.7~5.0 Damage Negligible, Felt by Most | 11 | 1,986 | 61% | 5 |
| 8 | Science Behind Idaho's 2020 Earthquake -us- | Mar. 31 2020 | Apr. 16 2020 | Mw 6.5 Fright General, Damage Slight | 16 | 7,135 | 59% | 16 |
| 9 | Science Behind Nevada's 2020 Earthquake -us- | May 15 2020 | May 29 2020 | Mw 6.5 Frightened All Damage Negligible | 14 | 4,252 | 57% | 13 |

| | | | | | | | | |
|---|---|---|---|---|---|---|---|---|
| 1 0 | Science Behind Mexico's 2020 Earthquake | Jun. 23 2020 | Jul. 05 2020 | Mw 7.4 Fright General, Considerable Damage | 12 | 1,420 | 60% | 1 |
| 1 1 | Science Behind the 2020 Sparta, North Carolina Earthquake -us- | Aug. 09 2020 | Aug. 25 2020 | Mw 5.2 Frightened All Considerable Damage | 16 | 4,147 | 65% | 10 |
| 1 2 | Science Behind the 2020 Aegean Sea Earthquake | Oct. 30 2020 | Nov. 16 2020 | Mw 7.0 Cracked Ground, Damage Serious | 17 | 2,732 | 57% | 7 |

**VEI:** Volcanic Eruptive Index (Global Volcanism Project, 2013).

**Mw:** Moment Magnitude Scale (Kanamori, 1977), the damage of the earthquake is described by the Modified Mercalli

Intensity (Wood and Neumann, 1931, Stover and Coffman, 1993).

**TIS:** Tsunami Intensity Scale (Papadopoulos, 2007).

(B) General GeoEd Videos:

| # | Year | Video Type | Video Example | Views * | Average View Percentage | Total Length | Comments |
|---|---|---|---|---|---|---|---|
| 1 | 2018 | Topical | Permian Basin Intro | 15,681 | 59% | 6min19sec | 12 |
| 2 | 2018 | Topical | What's happened inside Siberia's Mysterious Craters? | 1,958 | 51% | 4min24sec | 1 |
| 3 | 2018 | Topical | Nuclear Bomb and Radioactive Dating - Dating .. Wrong?? | 807 | 65% | 3min27sec | 2 |
| 4 | 2018 | Topical | Three Types of Igneous Rocks at Wichita Mountains | 1,329 | 54% | 5min2sec | 1 |
| 5 | 2018 | Topical | Why is the Moon white? | 7,425 | 48% | 3min54sec | 18 |
| 6 | 2018 | Topical | Evolution of the Permian Basin | 658 | 48% | 5min19sec | 0 |
| 7 | 2018 | Topical | Drilling to the Mantle | 1,905 | 64% | 3min21sec | 5 |
| 8 | 2020 | Topical | Are there volcanoes in Texas? | 23,191 | 60% | 5min33sec | 37 |
| 9 | 2020 | Simulation | Formation of a new subduction zone | 451 | 55% | 3min3sec | 1 |
| 10 | 2020 | Topical | What Happens When a Plane Flies into Volcanic Ash? | 1,984 | 67% | 2min33sec | 2 |
| 11 | 2020 | Basic Concept | The Four Types of Volcanoes | 23,617 | 52% | 2min45sec | 13 |
| 12 | 2020 | Topical | Induced Seismicity - The Oklahoma Story | 826 | 69% | 3min45sec | 1 |
| 13 | 2020 | Topical | Creatures of the Burgess Shale | 5,164 | 52% | 3min38sec | 13 |
| 14 | 2020 | Topical | Big Bend National Park | 1,095 | 77% | 3min1sec | 2 |
| 15 | 2020 | Topical | The Ogallala Aquifer | 8,563 | 55% | 4min20sec | 15 |
| 16 | 2020 | Basic Concept | Geodes: How Nature Creates Beautiful Mineral Formations | 3,300 | 60% | 3min16sec | 2 |

| 17 | 2020 | Video Abstract | Formation of a New Subduction Zone by Lithospheric Collapse around the Margins of a Large Plume Head | 423 | 54% | 3min15sec | 1 |
| 18 | 2020 | Basic Concept | How do Fossils Form? | 7,671 | 52% | 4min34sec | 7 |
| 19 | 2020 | Video Abstract | How Far South Might Himalayan Earthquakes Occur? | 2,345 | 52% | 4min26sec | 5 |
| 20 | 2020 | Basic Concept | Emergence: A chaotic system pushed into organization | 753 | 68% | 2min36sec | 4 |
| 21 | 2020 | Basic Concept | CO2 Drawdown - Where Should the Water Go? | 1,042 | 62% | 5min38sec | 7 |

## 5. Results

To analyze the six selected metrics, we first summarized the number of views of individual Geonews and General GeoEd videos (Table 2; Fig. 2A), as well as their performance after 1 and 3 years. Second, we compared the average views of both groups in the first 15 weeks after their release (Fig. 2B). Next, we compared the average viewed percentage of Geonews videos and General GeoEd videos over their lifetimes (Fig. 2C). Third, we summarized the differences of viewer age and gender for each group (Fig. 3 A and B). The ratio of like/dislike is reported in the text below. Lastly, we compared comments for both groups of videos (Fig. 4). These metrics are as of Oct. 3, 2021.

There are totally about 50,000 views of 12 Geonews video and ~110,000 views of 21 General GeoEd videos. The average number of views per video in 2018 and 2020 of General GeoEd videos (N=21) is 5,202 and that of Geonews (N=12) is 3,669. The standard deviation for General GeoEd group (SD=6,862) is much larger than that for the Geonews group (SD=1,650). The median views of Geonews videos is ~3,426, more than that of General GeoEd videos (1,958 views). The maximum views of General GeoEd and Geonews groups are 23,035 and 7,117 respectively, and the minimum views are 335 and 1,287 respectively. There are three General GeoEd videos with 15,000 to 25,000 views, which strongly influences the group mean and standard deviation (Table 2 and Fig. 2A).

Fig. 2A summarizes the number of views of videos released in 2018 (3-year lifetime) and 2020 (1-year lifetime) separately; data for each video is in Table 2. The mean of views for General GeoEd videos released in 2018 (~4,243) is greater than that of 2018 Geonews videos (~3,782). The standard deviation of 2018 General GeoEd videos is 5,126 while that of Geonews videos is 1,438. Moreover, for General GeoEd videos released in 2020, the average number of views is 5,681 (SD = 7,537). Geonews videos released in 2020, on the other hand, have a slightly smaller mean (3,613 views) and a much smaller standard deviation (1,744).

Second, to understand how the timeliness of Geonews videos affects viewer interest and how this differs from General GeoEd videos, we compared the weekly views of the two groups over the first 15 weeks after their release on YouTube (Fig. 2B). The results show that, on average, about 42% of total views of Geonews videos occurred in the first week after release (1,563 of 3,669). About 72% of views occurred in
the first two weeks (2,646 of 3,669) and approximately 78% in the first three weeks (2,880 of 3,669). Geonews group views in the first 15 weeks averages about 82% of the total (3,011 of 3,669). In comparison, General GeoEd videos average only 272 views in the first week of their release, only 5% of their total views. The number of first three-week views on average is 609 views, about 12% of the average total. In the first 15 weeks, General GeoEd group get 26% of the total views over their 1-3 year "lifetimes".
This difference is remarkable!

In addition to analyzing views, we compared the average length of views of both groups on YouTube (Fig. 2C). The average percentage viewed of Geonews video is 62±4%, which is slightly longer and more stable than that of General GeoEd videos (mean=58±8%). The maximum average percentage viewed of individual Geonews and General GeoEd videos is 68% and 76.5% respectively, and the minima are 57%
and 48%. The median average percentage viewed of Geonews videos is 61%, slightly higher than that of General GeoEd videos (55%).

Furthermore, to better understand the features of YouTube audiences of Geonews and General GeoEd videos, we studied viewer age and gender metrics (Fig. 3A and 3B). Most Geonews and General GeoEd viewers are above 65 years old (41.6% and 47.8%, respectively) but this may be partly skewed by the
280 demographics of the scientific societies where we advertise our videos (GSA, AGU, Sigma Xi, and AAAS). However, the second most important age group for the two video groups differ. Geonews videos got significantly more views from younger YouTube users. Young adults (25 to 44 years old) provide 36% of all viewers of Geonews videos, whereas the second biggest viewer group of General GeoEd videos are 45 to 64 years old. Both video groups got little interest from viewers younger than 25 years old (Geonews:
3.8% and General GeoEd: 4.3%). In terms of gender, most viewers of both video groups are male, but Geonews video viewers include more females. For Geonews videos, almost 20% of viewers are female compared to 10% for General GeoEd videos.  It is not possible to extract ethnicity information from YouTube data.

In addition, the ratio of like/dislike for Geonews videos is 98% (total like = 998, N=12) while that for
General GeoEd videos is 95% (total like =1968, N=21) by Oct 3, 2021. The small difference may not be significant.

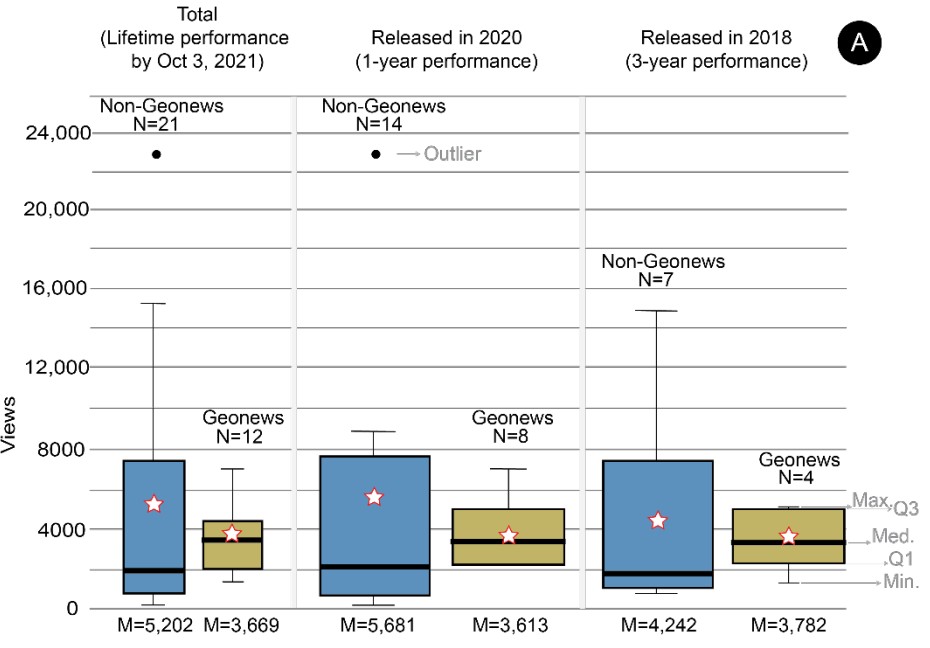

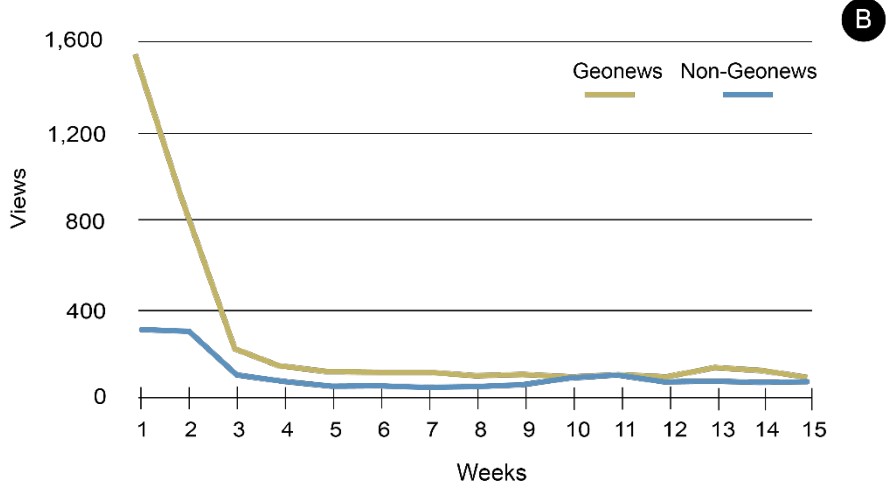

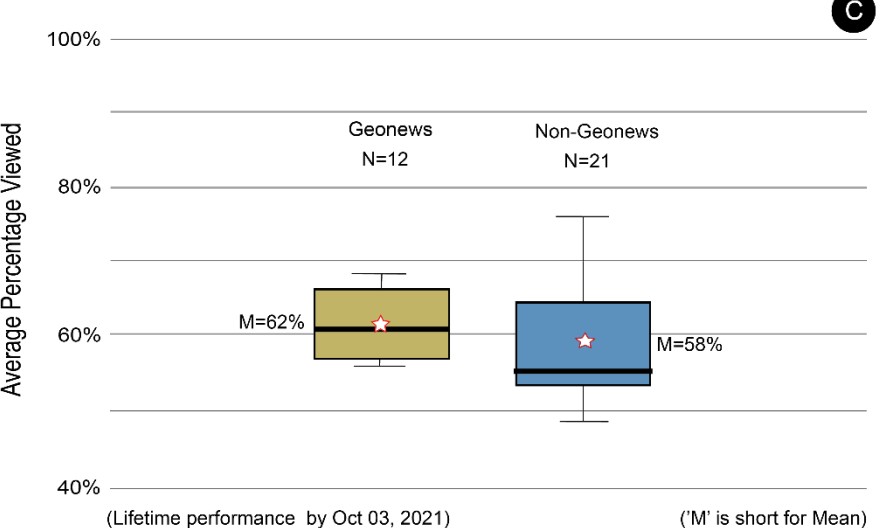

**Figure 2. Comparison of views and average percentage viewed of Geonews and General GeoEd videos. (A) Views of Geonews and General GeoEd videos in lifetime, 1 year and 3 years. (B) Average views of Geonews videos and General GeoEd videos over first 15 weeks following posting on YouTube. (C) Average view percentage of Geonews videos and General GeoEd videos.**

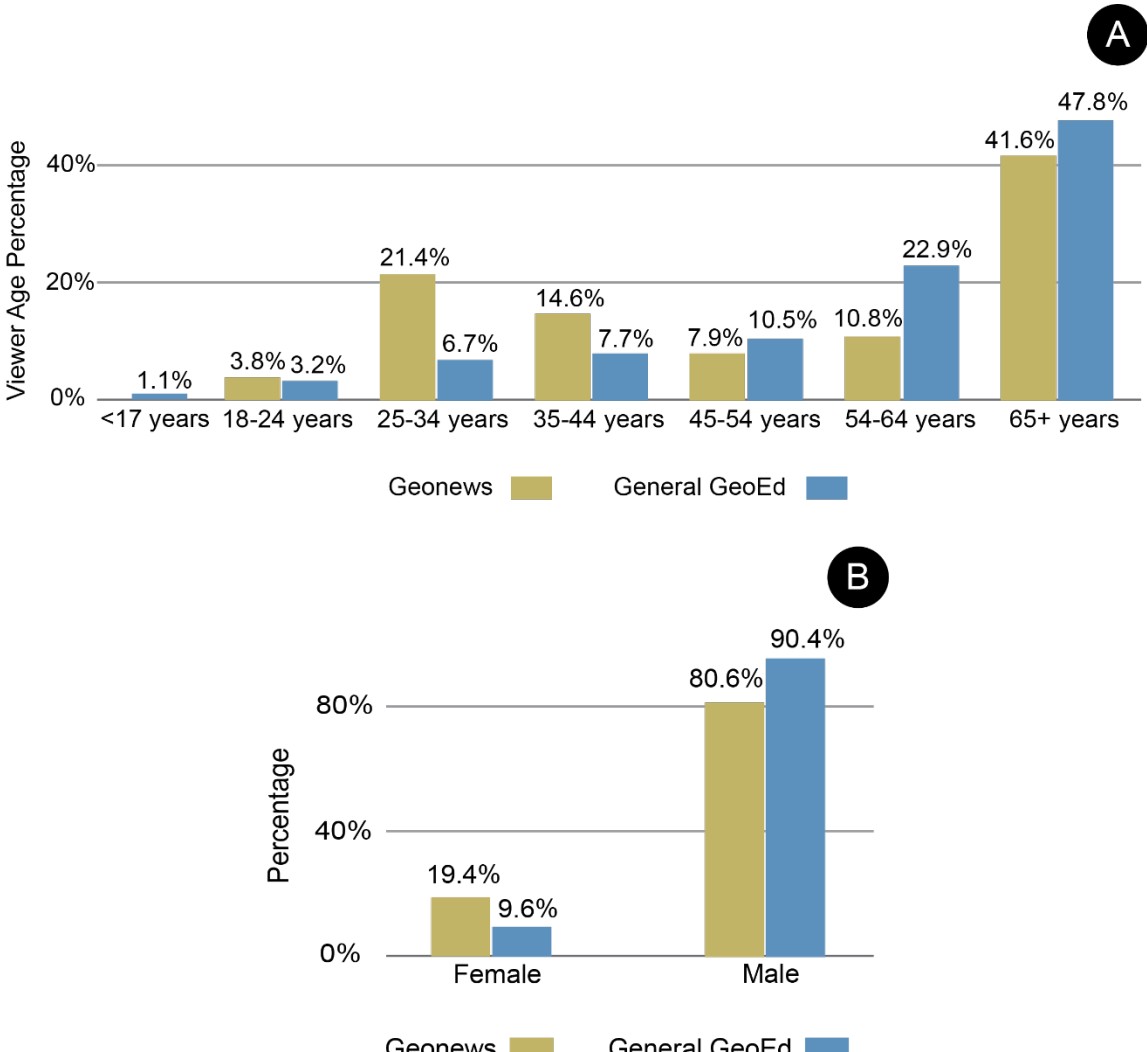

**Figure 3. Histogram of viewer ages (A) and gender (B) of Geonews and General GeoEd videos. The data is from 167,000 views of 33 YouTube videos by 10/03/2021 (~50,000 views of 12 Geonews video, ~110,000 views of 21 General GeoEd videos).**

Lastly, we summarized the comments (N=222) of Geonews and General GeoEd videos into 5 classes (Fig. 4): Meaningful dialogue, positive feedback, negative emotions, distrust, and other comments. From the past research of public understanding of science as well as learning engagement (Irwin and Michael, 2003; Michael, 2009; Dunn, 2013; Welbourne and Grant, 2016; Carmichael et al., 2018; Dubovi and Tabak, 2020), meaningful dialogue can involve personal experiences and observations (e.g. I live here and see.., I felt three quakes at home now I know why.., etc.), actively sharing relevant information, requesting more

information (e.g. references or more videos on relevant topics), giving advice for improvement (e.g. comments on video or audio quality; correcting pronunciations or clarify some terms), arguing about science, requesting to reuse videos for educational purposes. Positive feedback includes gratitude and applause for the video design. (Allum et al, 2008; Dubovi and Tabak, 2020). Negative comments show fear, anger or confusion (Allum et al, 2008). The distrust category expresses their distrust about news sources or biased conclusions due to funding sources. Other comments include advertisements, harassment, or irrelevant comments, etc. As of early October, 2021, there were 73 comments for Geonews videos (~6.1 comments/video on average, SD=~4.4) and 149 comments for General GeoEd videos (, ~7.1 comments/video on average, SD=~8.4). The number of comments for Geonews videos are more evenly distributed while General GeoEd videos have some with many comments (e.g. the General GeoEd video 'Are there volcanoes in Texas?' has 37 comments.). We found that more meaningful dialogue happened in response to Geonews videos than to General GeoEd videos (Fig. 4). Also, people who leave their comments under Geonews videos tend to share more about their personal experience and feelings, share more details, write longer comments (can be several paragraphs), and share their knowledge (such as the pronunciation of local names, what they know about the event, or time of the event, etc.).

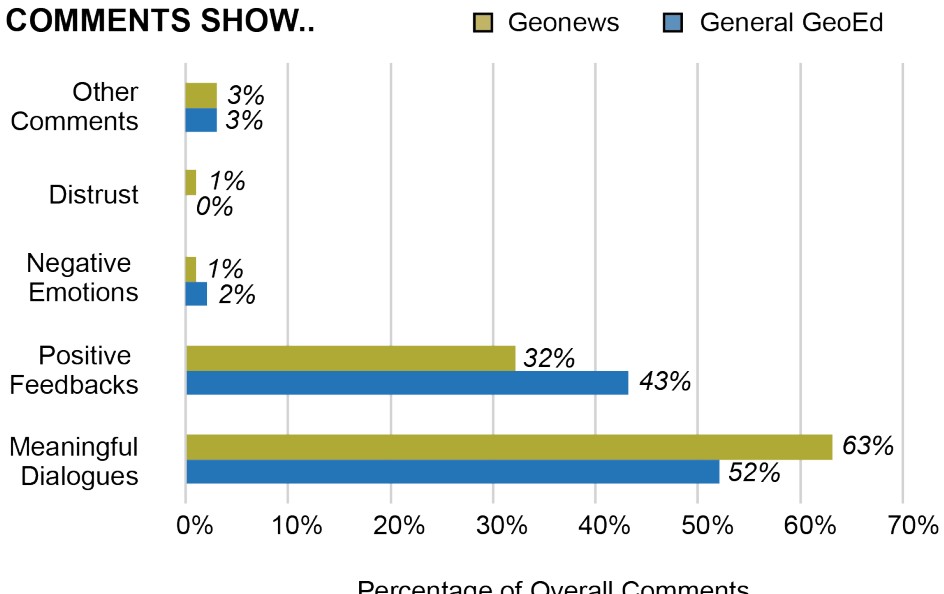

**Figure 4. Comparison of comments about Geonews videos (N=73) and General GeoEd videos (N=149). Datum as of 10/03/2021. All the values are rounded to the nearest integer. See text for detailed explanation.**

**6. Discussion**

To understand if and how timely natural hazard videos are useful for engaging YouTube viewers to learn more about Earth processes and communicate with geoscientists, we analyzed and compared six metrics of

Geonews and General GeoEd videos that we made and posted in 2018 and 2020. The results show that Geonews videos more consistently gain views compared to General GeoEd videos, which are much more variably attractive to the YouTube audience (Fig 2 and 3). In addition, Geonews videos have a slightly higher ratio of like/dislike than General GeoEd videos. These results indicate that the YouTube audience is interested in Geonews and the way it explains Earth processes. Geonews videos attracted audience more

steadily than General GeoEd videos, but some General GeoEd topics can be much more popular than Geonews videos. These data also indicate that Geonews videos may be useful in engaging younger and more diverse YouTube audiences than General GeoEd video, however, the potential of growth of views of the popular General GeoEd videos in the long-term is much higher than the Geonews videos (Fig 3).

       One result that is very clear is that most views of Geonews videos happen in the first few weeks after the

event (Figure 2B). About 82% of total views of Geonews videos occur within the first 3 weeks after release on YouTube, remarkably different from General GeoEd videos (12% of "lifetime" views in first 3 weeks). There is a big drop of views in Geonews videos after the initial 3 weeks; viewers are less likely to watch them after the 'golden period'. This may be related to audience interest but also can be influenced by the design of search engine or recommendation algorithm of YouTube. This needs further work to confirm.

Regardless of the reasons, our data shows that Geonews videos engage the YouTube audiences less after the first three weeks. Also, the 12 Geonews videos occurred in regions that include the USA, Mexico, Indonesia and Turkey-Greek areas. Viewers in these regions may be more interested in these videos than people living outside these regions. Moreover, the data shows that Geonews videos reach younger and more diverse audiences, at least in terms of gender, than do General GeoEd videos (Fig. 3). An important

demographic group that Geonews engaged better are YouTube users in the 25 to 44 years age old range. The more balanced gender and age distribution that Geonews videos attract reflects its potential to reach a younger and more diverse audience. It is hard to determine why higher percentage of younger and female users were reached by Geonews videos than the General GeoEd videos. We suspect it may be relevant to how different ages of people access to news. Younger generations may use YouTube as their major source

to watch news. The Pew statistics (2021) shows that 95% of US young adults (18 to 29 years old) routinely use YouTube (Statista, 2020). The time that young adults spend on YouTube has increased continuously over the past few years (Kaul et al., 2020). Survey results from Wissenschaft (2018) for Germany shows that 42% of 14- to 29-year-olds use YouTube frequently or very frequently to inform themselves about science. This evidence shows that YouTube plays an increasingly important role in the learning habits of

today's young people (Boy et al., 2020). Kaul et al. (2020) argued that if environmental science communicators are serious in their efforts to reach young people, new strategies based on YouTube need to be devised. The results of this study support these conclusions. Nearly half of the audience for Geonews YouTube videos viewers are young to mid-life adults (ages 19-44 account for about 48% of total viewers).

       In addition, our analysis of comments shows that meaningful dialogue occurred more often with Geonews

videos (63%) than with General GeoEd videos (52%) (Fig. 4). Although the data in this work is limited

(222 comments from 160k views) and the commenting audience members may not be representative of their communities (see Sec. 7 Limitations), we see users living near the event leaving comments on about half of the Geonews videos in this study (even Geonews videos with fewer views, e.g. Mentone, TX earthquake and Aegean Sea Earthquake). These comments involve feelings, thoughts, experiences and lay knowledge about the events. From analyzing these comments, we tentatively conclude that people living in the region affected by the event are more likely to leave comments on Geonews videos. A possible explain for this may be related to the difference between the "Publics-in-General" and "Publics-in-Particular" (Michael, 2009) as well as the high level of the 'lay local' knowledge of viewers who live in the affected region (Allum et al., 2008). Research shows that when the public tries to understand science, they also regard themselves as one of these "publics" (Irwin and Michael, 2003; Lacchia et al., 2020). Local people may think that a nearby event differentiates them from others because they know more about it as well as being more affected by it. Such 'lay local' knowledge may increase their willingness, confidence and motivation to share and communicate on YouTube (Dunn, 2013; Welbourne and Grant, 2016; Carmichael et al., 2018; Dubovi and Tabak, 2020). This may be responsible for the higher possibility of having longer and more detailed descriptions of their personal experiences under Geonews videos. Additional evidence supporting this hypothesis is that most comments on Geonews videos concern the event rather than about video design which comprise a larger proportion of comments on General GeoEd videos. This tendency of people in the affected region to want to share personal thoughts and experiences about a timely event has been observed for Twitter and Facebook (e.g. Lacassin et al., 2020; Hugelius et al., 2017). We discuss the differences of comments among YouTube, Twitter and Facebook in later sections.

### 6.1. How reliable are the YouTube Analytics data and is it ethical to use the data?

The reliability of YouTube metrics data is largely determined by how YouTube (and its parent company Google) gets the data. The video watching and channel metrics, such as the number of views, are collected via the YouTube platform. The data is relatively accurate, especially considering the magnitude of the data and partially reflects YouTube efforts to correct these (Talreja, 2021). Some concern is given to the reliability of gender and age data. When users register a Google account, they are asked for basic demographic information such as name, age, and gender. Since there is no way to verify the accuracy of this information, users could provide false information. User information is available via YouTube Analytics to those logged into Google services including Google Chrome Browser and YouTube. Google will also predict users' age and gender utilizing advertisement clicking behaviors and cookies. Google does not publish the accuracy of their age or gender data, so we can only discuss its accuracy from indirect evidence. First, some studies used demographic data from YouTube to train models predict the users' demographic features, with good results (e.g. Ulges et al, 2013). Second, Tschantz et al (2018) did a survey-based research on the accuracy of Google age and gender data inferences and concluded that Google accurately estimates the data. Therefore, considering the magnitude and period of the data collection, the population nature of the dataset (not samples), we suggest that the results we got from the YouTube Analytics data in this study are reasonably reliable.

Based on past discussions of social media research ethics (Association of Internet Researchers, 2012; Townsend and Wallace, 2016; Woodfield, 2017; Golder et al, 2017; Legewie and Nassauer, 2018), the ethics of Geonews project using YouTube Analytics data and comments content analysis are considered in three parts: (1) Informed consent; (2) If the data are public or private; and (3) Is there any potential risk? Informed consents were collected from users when they register for their Google accounts. Although many argue that the consent is just a box to tick in the terms and conditions (e.g. Nature Editorial, 2019), we argue that this consent is adequate for our study since it is a minimal risk project. We use data that are either completely anonymized and aggregated or are voluntarily posted by YouTube users as comments for public view. The risk of harm for using and reporting these data is minor. For these reasons we think that using this data in this study, although without specifically informed consent for our study, is ethical.

## 6.2. How do comments differ between Twitter, Facebook and YouTube?

Social media platforms encourage participation, sharing, interaction and collaboration using online technologies, but have different styles and foci (Pavelle and Wikinson, 2020). Common types of social media include blogs and microblogs (e.g. Twitter), content communities (e.g. YouTube), and social networking (e.g. Facebook). Some argue that because YouTube is limited to video content (Zuckerberg et al., 2012), most of the comment threads and discussions can be ignored by other users who are interested in the videos. It is true that most discussion threads on YouTube are not as detailed as those on Twitter or Facebook and that posting rates are also relatively low (Moran et al., 2011). Users who leave comments on YouTube videos may not expect feedback from other YouTube viewers but they may ask questions to the video uploaders. This is seen in our study too. Therefore, scientists posting YouTube videos are encouraged to pay more attention to answering YouTube comments because it is possibly to establish emotional and mental connections in this way (Pavelle and Wikinson, 2020; Smith, 2020).

## 6.3. How are videos and Geonews videos found on YouTube?

We advertise our videos via on-line communities of three scientific societies: The Geological Society of America, the American Geophysical Union, and Sigma Xi. These audiences are older and more knowledgeable about Earth processes than the general public. We advertise our videos to the general public using what YouTube offers. In general, YouTube videos can be found by two ways, search and recommendations (Landrum et al., 2021). Search results are largely determined by videos' relevance, historical views and likes (Zhou et al., 2010). On the other hand, the YouTube recommendation system adopts machine learning models (Covington et al., 2016; Beautemps and Bresges, 2021). There are several special features of machine learning models that relevant. First, the models consider the upload time and time-dependant popularity; Geonews videos benefit from this feature. Second, the models try to match user language and video language. This may explain why Geonews videos outside US get fewer views, even though some events are important (e.g. Mexico earthquake 2020 or Aegean Sea earthquake 2020). Third, the watching time and percentage of views are important factors reflecting engagement in the YouTube

recommendation models. Therefore, the higher average percentage of views Geonews videos may also make them more recommended than general GeoEd videos.

Aside from YouTube's video searching and recommendation system, the popularity of a video also depends on its content and content-agnostic factors (Borghol et al., 2012; Figueiredo et al., 2014; Velho and Barata, 2020). Content factors include the stylistic and informational characteristics of a video (e.g. thumbnail, topic, design). Content-agnostic factors reflect the popularity of the creator or partner's social network or video upload date and time (Khan and Vong, 2014). One content-agnostic strategy is to join with YouTube influencers to help promote videos (Geipel, 2018; Nafees et al., 2021) but the results for individual projects may vary (Donhauser and Beck, 2021). Research also shows that, compared to the YouTube algorithm and content-agnostic factors, content factors are most influential for the popularity of a science video (Figueiredo et al., 2014).

Geonews videos are designed to catch the momentum of timely natural hazards to engage the public. Therefore, we expected that the views of Geonews videos would correlate with timeliness of video after the event. However, no significant relationship between release speed and views is found (R = 0.12, with $R^2$=0.015), which is unexpected. At present, our team needs about 2 weeks (4-18 days; mean = 13.5 days) to create a Geonews video (Table 2A). The most popular videos are posted within a week after the event. We suspect that our release speed is too slow to catch viewers' peak interest and that a faster release after the event would receive more views.

Also, the popularity of Geonews videos seems to be influenced by geography. YouTube provides some geographic data for videos but 50 - 95 % of the geographic data for where viewers are is missing or inaccessible. Thus, we do not have enough data to conduct a robust investigation of the geographic distribution of audiences for each Geonews video. However, our results (Table 2) show that 5 of the 6 most viewed Geonews videos (>4,000 views) are US events. Events of other Geonews videos occurred in Indonesia, Philippines, Turkey-Greece, and Mexico, with native languages that are not English. Thus, we suspect that a geographic feature of Geonews audiences may be at least partially related to the language feature of the search and recommendation algorithms used by YouTube as discussed in previous paragraphs. Also, although we add English closed-captions, non-English speakers probably have great difficulty to follow the Geonews videos. This reinforces the needs of having multiple language versions of Geonews videos, and encourages local geoscience teams to create Geonews type videos to engage local audiences.

Lastly, we expect that the significance and type of the events will affect the popularity of Geonews videos. Although the significance of an event to the public is related to damage and casualties as well as the magnitude of the event, as well as the population affected by the event it is still hard to compare the significance in the public mind of different types of geohazard events. Thus, the results of this work are not enough to estimate the correlation between significance of an event and the popularity of the Geonews video about it. However, we conclude from Table 2 that more destructive and powerful events near US

population centers will be most popular. Due to our limited videos for each type of geohazard, we cannot tell what types of geohazards are more popular for what audiences (e.g. previous experience of hazards or geographic distributions), but this can be an interesting future research direction.

## 7. Limitations

A major limitation of our method is that the number of assessed videos is restricted to those posted on the
UTD GSS YouTube channel (with about 2,500 subscribers by Feb. 2022). The effect of channel popularity is not tested in this research. More popular channels (such as NASA) and smaller and less popular channels (such as new channels with very few subscribers) may have different results if they undertook a similar experiment. However, we are unaware of any other YouTube channel that makes a range of GeoEd videos comparable to those of UTD Geoscience Studios and also makes something like Geonews videos (IRIS
recently started a new channel and released some Geonews-like videos, named 'IRIS Teachable Moments', but it is separate from their major channel. We have no access to the data for individual videos, therefore, we did not incorporate this in our analysis.) In addition, although the General GeoEd videos have various designs and topics, the number of General GeoEd videos as a control group may not adequately capture YouTube audience interest. However, with a combined method of quantitative and qualitative ways to
assess YouTube video design elements, the results provide useful insights into the engagement potential of short, timely videos about natural hazard events in the news as an important element of GeoEd videos. Furthermore, the emotional impact of Geonews videos is another concern. Timely information about hazards may trigger fear, anger, distrust and other negative attitudes and feelings. This is seen in about 2% of the YouTube comments. Video makers may need to use more time to reply to comments and share more
information in an effort to respond to negative comments (Takahashi et al., 2015; Jones, 2020; Lacassin et al., 2020). It may be useful to share some resilience knowledge (Van Loon et al., 2020) or hazard simulation games (e.g. Kerlow et al., 2020) to help these viewers.

Another limitation is that there are few comments considering the views (222 comments for 160k views, ~0.1% comment rate) and the numbers of comments for each video varies (0 to 37 comments). It is hard to
argue that the comments on the videos are representative of the viewing audience. As discussed above, we suspect that the audience near the event may be especially motivated to leave comments about their personal experiences or about the events. A more in-depth method (survey or interview of commented audiences) is needed to better understand audience motivations, which is an interesting future research topic.

## 8. Conclusions

Our study shows that timely videos about Earth events in the news are useful for engaging the public and show promise for reaching younger and more diverse audiences. Results of this research suggests that

short, timely videos about natural hazards and events especially engage people who live near where it occurs, motivating them to learn and discuss the geoscience behind these events. Although Geonews videos

might have fewer total views than some popular General GeoEd videos, Geonews videos are especially good at starting meaningful dialogue and engage YouTube audiences for several weeks after the event happens. The popularity of Geonews videos has a geographic aspect that can be enhanced by adding pertinent languages.  We encourage others to add captions or voice-over to any of our posted videos. There are opportunities for geoscientists around the world to create Geonews videos focusing on regional events

using local languages as well as translating Geonews videos. Moreover, considering the production efficiency compared to other GeoEd videos, engaging audiences with Geonews videos on YouTube is a very promising strategy. Geoscientists can create YouTube Geonews videos to partially fulfill their needs of delivering scientific information, but taking time to reply to YouTube comments (not only Geonews but all kinds of GeoEd videos) could also be important for meaningfully communicating topical geoscience to

the public (just like some scientists do with Twitter, e.g. Lacassin et al., 2020; Pavelle and Wilkinson, 2020).  Our findings about Geonews videos may encourage other types of timely event-based educational videos as well.

**Acknowledgements:**

We would like to thank the editors Prof. Mirjam Glessmer and reviewers of Geoscience Communication Prof. Joachim Allgaier and Prof. Robin Lacassin. Their insightful comments and suggestions greatly helped the completeness of this work and improved the quality of the paper. Thanks to assistance from Siloa Willis, Kathryn Creecy, Clinton Crowley, Katie Seals, and Lochlan Vaughn. This work was partially supported by NSF grant 1712495 and a grant from the AAPG Foundation. This is UTD Geoscience
contribution # 16xx.

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
