# Peer review of "Geonews: Timely Geoscience Educational YouTube Videos about Recent Geologic Events"

_Geoscience Communication, 2021_

## Author Response (AR1)

GEOSCIENCES

**THE UNIVERSITY OF TEXAS AT DALLAS**

FO21   P.O. BOX 830688   RICHARDSON, TEXAS 75083-0688
(972) 883-2401     FAX  (972) 883-2537

Feb.  8, 2022

Dear Geoscience Communication Editors,

We hereby submit the minor-revised manuscript for consideration of publication in the Geoscience Communication: **Geonews: Timely Geoscience Educational Videos about Recent Geologic Events.**

This paper explores the usefulness of making timely events-based videos to better engage with and communicate geoscience to the public on YouTube. We applied a video-making based method and analyzed the impact of 33 short geoscience educational (GeoEd) videos that we created and posted on YouTube in 2018 and 2020. These short GeoEd videos are either Geonews videos (about timely geologic events) or non-Geonews videos (about geology but not about timely events). By comparing the performance of Geonews and non-Geonews videos quantitively and analyzing the comments qualitatively, we discovered how short, timely videos about geologic events in the news better engages younger and more diverse audiences to be interested in the Earth Science and stimulates more meaningful dialogues with the public. We also share workflow and design principles of Geonews videos, which should be useful for other scientists who want to use YouTube to communicate timely events. We think this paper will be of broad interest to the readers of your journal.

The paper was submitted to another journal, 'Public Understanding of Science' on May 2021 and was reviewed by two reviewers and was rejected on July 2021. The reviewers gave very useful advice. Later, after reviewing by two other reviewers (Dr. Joachim Allagier and Dr. Robin Lacassin), this 2022-02-08 version of the manuscript was revised based on their comments updated. The details of their comments and our responses are at the end of this cover letter.

Potential reviewers:

Dr. Joachim Allgaier < joachim.allgaier@humtec.rwth-aachen.de>

Dr. Robin Lacassin < lacassin@ipgp.fr >

Dr. Jill Karsten  <jkarsten55@gmail.com>

Dr. Kim A. Kastens < kastens@ldeo.columbia.edu >

Dr. Johanna Ickert<johanna.ickert@plymouth.ac.uk>

Dr. Will J. Grant < will.grant@anu.edu.au>

Dr. Wendy Bohon <wendy.bohon@iris.edu>

Sincerely,

Ning Wang, Zachary Clowdus, Alessandra Sealander and Robert J Stern

REVIEWER #1

This is an interesting manuscript. The situation is that more and more people worldwide use YouTube as an information source (also about science and research) and therefore it is very important that scientific and other experts are present on the platform as well and provide good quality information. In this sense, the manuscript is a valuable starting point for the geoscience community to exchange information and knowledge about how this goal can be attained.

I think it is a big plus of the manuscript that it also engages with the public understanding of science / public engagement literature. The approach presented is convincing and the results seem plausible. I think it is not surprising that timely news videos are watched mostly close to a specific event and more foundational videos are watched and are still interesting over a longer time-scale. Both are helpful tools to engage and inform general audiences.

**NING: Thank you so much for so many encouraging words! Greatly appreciate it.**

However, here I think it might be worth thinking a bit about how videos are actually found on YouTube and what could be possibly done to improve the probability of the videos being found (e.g. use of particular hashtags, a twitter account for the promotion of the videos, promotion of the videos in the comment section of popular science youtubers, cooperation and networking with other education / science channels etc.). That might be a topic of the discussion section or for further research.

**NING: Yes, thanks for your great idea, we have added a section to discuss how videos can be found on YouTube and possible ways to enhance video vitality on YouTube, as well as some discussions about the advantage of Geonews videos being vital. Please see the '6.3. How are videos found on YouTube and what are the advantage of Geonews videos? ' Starting at Line 425.**

Please double check the references. For instance, I did not find the references Allgaier (2019) or Tao et al (2014) in the list of references despite them being cited in the text – more might be missing.

**NING: All references are fixed and checked! Thanks!**

As a general note, it might be of interest, that Asheley Landrum and I have curated a research topic on the use of online video formats in science and environmental communication. Some of the contributions might be of interest to the authors of the manuscript. They can be found here:
https://www.frontiersin.org/research-topics/11604/new-directions-in-science-andenvironmental-communication-understanding-the-role-of-online-video-sha
I personally think that it is great that you take the time and effort to make these videos. It is actually a very valuable service provided to the general public.
Best wishes
Joachim Allgaier

**NING: Appreciate it! Prof. Allgaier. These works greatly developed my knowledge and inspired my research. Thank you for your encouraging words and recommendations!**

REVIEWER #2:

Ning Wang et al.'s paper evaluates the way YouTube videos about timely geological events (Geonews videos) and about general geological topics (GeoEd videos) are viewed and received by the public. They aim to explore questions about public interest. To do that they used a relatively limited dataset (12 Geonews, 21 GeoEd) made of videos they have created and posted on YouTube between 2018 and 2020. It's an interesting subject that deserves publishing in Geoscience Communication. I found the study interesting and useful but perhaps not complete, with unaddressed possible biases, and with some aspects needing more explanation. I list below some comments and suggestions with the aim to broaden and strengthen the presented analyses and discussion. Hope this will help.

**NING: Thank you so much! Your insightful comments are very helpful and lead us to some interesting considerations and discussions. Making our discussion more completed. Appreciate it.**

General comments:

It's interesting to keep in mind that all these videos have been created by the Geoscience Studio at the University of Texas at Dallas, USA, and that there is no other dataset to put author's analyses and conclusions in a broader perspective. Among their 12 Geonews, 6 are about US events, plus one in Mexico, and 5 about events far from the US; 8 are about earthquakes, 4 on volcanic eruptions. I note that five of the six most viewed (>4000 views) Geonews videos are about US events. There is a clear geographical correlation. This may perhaps introduce biases in the interpretation when looking to other factors, and rises the questions of correlation / independence of those factors. When looking to GeoEd videos, it's little more complicated, some US subjects got a lot of views (but not all), but most viewed videos are about fossils plus a general one on volcano types. I suggest to add some qualitative words about that, and better discuss these issues concerning video's geographical location and/or topics.

**NING: Very interesting point, we have added two paragraphs in '6.3 How are videos found on YouTube and what are the advantage of Geonews videos?' to specifically discuss the geographic and geohazard feature of videos. (please see Line 455 to 475).**

Some of the presented analyses rest on attributes that describe the public (age, gender…), taken from YouTube data. It's for me a total "mystery" how these attributes are determined as most YouTube viewers are likely anonymous / unconnected / with a fancy user name / etc... I infer it's from google analyses of user's data based on login name (when logged), IP number, cookies, or other blackboxes. This needs explanations. You mention and use age and gender data. How reliable are these data? Please explain how these data are obtained, the uncertainties, possible biases, etc. How can this affect your conclusions? Is it deontological fair to use data on which you haven't any control and that you need to trust? Is it ethical (even if the data are completely anonymized)? This deserves discussion.

**NING: Thank you for asking these questions, we have added a section '6.1 How reliable are the YouTube Analytics data and is it ethical to use the data?' to specifically discuss the geographic and geohazard feature of videos. (please see Line 385 to 415)**

There are few comments posted on the videos (a total of 222 comments for all Geonews /GeoEd videos) compared to the number of views (several 10k views). An important question is thus: Is your analysis meaningful and representative of the public, or mostly anecdotic? How does it help to better shape future videos? Are these results useful and why? More generally, is this small percentage of people engaging discussion the same with other social medias, like facebook feeds for example?

**NING: Thank you for asking these questions, we have revised the discussion section and added a section '6.2. How comments are different among Twitter, Facebook and YouTube?' to discuss the difference of comments on these three social media platforms. (please see Line 415 to 425)**

I also have some concerns about your analysis of the viewing percentage versus video length. See below.

I haven't detected other issues on other sections of the analysis and discussion.

**NING: We agree, the relationship is not significant and didn't add much value to the discussion. Thus, we deleted the 'viewing percentage versus video length' part.**

More specific comments:

• Tables 1 and 2 (pages 9 to 11):

- Why are you listing the videos in reverse releasing time (more recent first, but

numbered #1)? This complicates evaluation.

**NING: Good point, thanks. The videos are listed in chronological order now.**

- Strictly speaking this is not "event time", "release time", but calendar dates. And the

format you are using may be confusing (exemple: 07 05 2020 = month first? or day?),

please clarify.

**NING: Yes, sorry for the confusing format, we have revised the format and the term in the new version of the manuscript.**

- What is the meaning of the asterisks on table 1 (TYPE*) and table 2 (table title,

VIEWS*)?

**NING: * indicates the date of the statistics, as of Oct 03, 21. We clarified this after the Table 1 and 2A**

- I would recommend rounding the values of the percentages, and providing integer

values. - I would like to know how many persons viewed 100% of each video (is it

available from YouTube data?).

**NING: Unfortunately, this data is not accessible. But it is an interesting angle, thanks!**

- I suggest to add a column to help identify US related events / subjects (see comments

above).

- Add a column with the number of comments for each event

**NING: Yes, we have added them.**

• Section 7 of the manuscript (lines 410-420 and related Figure 5):

- In this section you claim that there is a "strong negative relationship between video length and percentage viewed for Geonews videos" (but less clear for GeoEd ones), meaning that "shorter videos are more engaging". However, looking to Figure 5, I found this negative correlation unconvincing even if it's numerically right with your dataset. You are working with a very small number of events; adding a single captivating video of about 5 minutes would flatten or even reverse the line. IMO this is not good evidence for a "strong negative relationship". I would rather say that all Geonews videos got between 57 and 68% average view percentage, thus not very different from one to another. And I would perhaps argue for a

"loose possible relationship" btw length and that percentage, and without giving much significance to these results. Rather than an average percentage it would be more interesting to know how many people viewed 100% of each video, and what is the viewing percentage distribution (I don't know if you can get this from youtube data).

**NING: We agree, the relationship is not significant and didn't add much value to the discussion. Thus, we deleted the section. Interesting idea about the audience analysis, who finish the video and for each video, we can only have viewing percentage distribution.**

- I note that one data point is missing on the diagram of Figure 5 (only 11 are plotted). I don't know if you used it in your correlation or not.

- I also note that Figure 5 is not cited in the text.

**NING: Figure 5 and corresponding paragraph is deleted.**

- I again recommend rounding percentage values to integer values (in the tables).

Robin Lacassin / Paris / January 2022

**NING: Very Good Point, no meaning to complicate the number, have already revised in the paper.**

Comments to the author:
Dear Ning and co-authors,

I apologise for my delayed response.

You have received very helpful comments from the two reviewers and I am happy to recommend this manuscript for publication when you have addressed a couple of points:
- I would like to see a discussion of how the videos are found (for more helpful questions, see RC1). This will increase the manuscript's usefulness to readers as they understand better what works and what does not. It might also help fine-tune your own practice.
- Please include discussions of RC2s general comments
- As was pointed out by both reviewers, there are some formatting issues with the manuscript, e.g. references in the text and bibliography not being consistent, a figure not being referenced in the text. Please double-check and correct!

Best regards, Mirjam Glessmer

**NING: Thank you so much for your comments and review efforts. We have revised all the points that reviewers mentioned and fixed all the formatting issues. The details are above.**